# Nest Turrets of *Acromyrmex* Grass-Cutting Ants: Micromorphology Reveals Building Techniques and Construction Dynamics

**DOI:** 10.3390/insects11020140

**Published:** 2020-02-24

**Authors:** Marcela I. Cosarinsky, Daniela Römer, Flavio Roces

**Affiliations:** 1Departamento de Ciencias Geológicas, Facultad de Ciencias Exactas y Naturales, Universidad de Buenos Aires, Pabellón II, Intendente Güiraldes 2160, CP C1428EHA, Ciudad Universitaria, Buenos Aires B1657, Argentina; mar28c@hotmail.com; 2Unidad de Entomología, Departamento de Protección Vegetal, Facultad de Agronomía, Universidad de la República, Av. E. Garzón 780, Montevideo 11200, Uruguay; Daniela.Roemer@uni-wuerzburg.de; 3Department of Behavioral Physiology and Sociobiology, Biocenter, University of Würzburg, Am Hubland, 97074 Würzburg, Germany

**Keywords:** building behavior, *Acromyrmex fracticornis*, leaf-cutting ants, collective pattern, architecture, self-organization, microstructure, material composition, thin sections

## Abstract

*Acromyrmex fracticornis* grass-cutting ants construct conspicuous chimney-shaped nest turrets made of intermeshed grass fragments. We asked whether turrets are constructed by merely piling up nearby materials around the entrance, or whether ants incorporate different materials as the turret develops. By removing the original nest turrets and following their rebuilding process over three consecutive days, age-dependent changes in wall morphology and inner lining fabrics were characterized. Micromorphological descriptions based on thin sections of turret walls revealed the building behaviors involved. Ants started by collecting nearby twigs and dry grass fragments that are piled up around the nest entrance. Several large fragments held the structure like beams. As a net-like structure grew, soil pellets were placed in between the intermeshed plant fragments from the turret base to the top, reinforcing the structure. Concomitantly, the turret inner wall was lined with soil pellets, starting from the base. Therefore, the consolidation of the turret occurred both over time and from its base upwards. It is argued that nest turrets do not simply arise by the arbitrary deposition of nearby materials, and that workers selectively incorporate large materials at the beginning, and respond to the developing structure by reinforcing the intermeshed plant fragments over time.

## 1. Introduction

Ant colonies inhabiting underground nests often build a conspicuous mound by depositing the excavated soil and additional material collected from the surroundings around the nest entrances. The nest mounds may have either a compact crusted or a soft thatched surface, and a more porous inside structure usually permeated by galleries and cavities, with a texture, chemistry, and microstructure that greatly differs from the neighboring soil [1,2,3].

Neighboring ant colonies of different species inhabiting similar soils construct nests that markedly differ in their morphology, indicating that the final nest structure does not necessarily depend on the available building materials. In the humid grasslands of South America, for instance, colonies of the ant *Camponotus punctulatus* (Formicinae) construct distinct soil mounds locally called “tacurúes” in low, frequently flooded areas of the Chaco region. They are cone-shaped, 1–2 m high, extremely hard, and frequently covered with vegetation [4,5,6]. *Camponotus* “tacurúes” are often observed like islands surrounded by water in flooded areas of North Argentina, with the colony inhabiting the uppermost nest part (personal observations by Marcela I. Cosarinsky), suggesting that mound building behavior was selected for as an adaptation to survive in frequently flooded soils. In contrast, fire ant colonies (genus *Solenopsis*, Myrmicinae) occurring in the same habitat construct smaller mounds composed of loose soil that can easily be mechanically damaged [5,7,8].

Leaf-cutting ants, particularly those of the genus *Atta*, excavate the largest and most complex nests among ants [9,10,11,12,13]. Colonies house developing fungus gardens in a large number of underground chambers, and workers dig in addition voluminous chambers for the deposition of colony waste. Nest-building behavior has been studied in detail in the grass-cutting ant *Atta vollenweideri* [14,15,16,17], the colonies of which construct large mounds of up to 8 m in basal diameter and 0.8 m in height. A mature mound is permeated by ca. 100 openings that lead, via an intricate tunneling system, to the underground chambers. The peripheral nest openings are used as nest entrances and connect to foraging trails, while the central ones are on top of conspicuous turrets that foster air exchanges between the nest interior and the environment [9,18,19,20,21].

Leaf-cutting ant colonies of the genus *Acromyrmex* are comparatively smaller than *Atta* colonies, and within the genus a variety of underground, epigean, and even arboreal nesting habits occur [22]. In some species, the nest mound results from the simple accumulation of excavated soil around the entrance, while in others, the mound is a thatched structure composed of plant material and soil particles, as in *Acromyrmex heyeri*, *Acromyrmex hispidus*, and *Acromyrmex lobicornis* [7,23,24,25,26,27,28,29]. Besides mounds, colonies of three closely-related species of *Acromyrmex* grass-cutting ants, *Acromyrmex landolti*, *Acromyrmex balzani*, and *Acromyrmex fracticornis*, construct conspicuous chimney-shaped turrets around the nest entrance made of intermeshed grass fragments and soil particles [30,31,32,33,34]. It has experimentally been demonstrated that turret construction in *A. landolti* mostly occurs in the wet season, and that the turret walls are very resistant to water infiltration, suggesting that turrets protect the colony against flooding [35]. The construction of “levees” around the nest entrance is not only limited to grass-cutting ants. Workers of *Ectatomma opaciventre* construct solid turrets with intermeshed plant fragments and soil that hold back water and prevent nest flooding [36,37], while in *Dorymyrmex thorasicus* and *Pheidole obscurithorax*, less consolidated structures are built, which are more susceptible to collapse by heavy rains [37].

While only a few investigations focused on the function of nest turrets in *Acromyrmex* grass-cutting ants, as indicated above [31,35,37], no single study investigated how workers assemble the building materials, namely grass fragments and soil particles, to construct the turret. In the present study, we addressed the question whether turrets built by *A. fracticornis* are constructed by merely piling up nearby materials around the nest entrance, or whether ants respond to cues provided by the turret being constructed and incorporate different materials to the structure as it develops. To highlight the building techniques used by ants, we investigated turret construction in field nests located in Formosa, Argentina, by removing the original nest turrets, and by following the rebuilding process over three consecutive days until the new turret was finished. By means of thin sections and micromorphological analysis of the turret walls, age-dependent changes in wall morphology and inner lining fabrics of developing turrets could be characterized and compared with those of the original turrets, to reveal the behaviors involved in turret construction.

## 2. Materials and Methods

### 2.1. Area Description

This study was performed at the Reserva Ecológica “El Bagual” (26°18’18.4’’ S, 57°49’51.0’’ W), located in the eastern humid Chaco region, near San Francisco de Laishi, Formosa province, Argentina, during two rainy seasons (February–March 2015, and March–April 2017). The field station is privately owned by the Estancia EL BAGUAL-ALPARAMIS S.A., and research was conducted with permission of the owner Pablo Götz and the station supervisor Alejandro G. Di Giacomo. The studied species, *Acromyrmex fracticornis*, is not protected under the Convention of International Trade in Endangered Species (CITES).

### 2.2. Sampling, Morphological, and Micromorphological Analysis

Nest turrets and their surrounding soil were sampled along a field path covered with natural grass plants. Nests were very common in the area, and samples were collected from nests located at least 2 m apart. Based on morphological data, the soil was determined as an entic Hapludoll (Mollisol), with an A horizon (0–30 cm, grayish brown), an AC horizon (30–60 cm, reddish brown), and a C horizon (60–100 cm and beyond, dull reddish brown).

To quantify the dimensions of a typical nest turret, 40 nests with a turret having a single opening (some turrets may have two or more adjacent openings; see Section 3.1.) were selected, and their turrets carefully removed with a small garden shovel. Turret height, the maximal and the perpendicular diameter at the turret base, the diameter at the turret top, and the opening diameter were measured to the nearest 1 mm. To investigate their material composition, five additional nest turrets were collected, dissolved in demineralized water, their components separated into plant and soil materials, and both materials separately weighed to the nearest 0.1 g after drying overnight at 50 °C.

To identify the building techniques used by ants over time to construct their nest turrets, we selected and numbered 57 nests of *A. fracticornis* having turrets with a single opening. We took photos of each nest turret and then carefully removed the turrets from the soil at their base with a box cutter. To follow the process of turret rebuilding, we sampled newly constructed turrets after either one, two, or three consecutive days until the new turret was finished, since no distinct morphological changes were apparent three days after their reconstruction. For comparisons, the surrounding soil (A horizon) was also sampled at 10 cm depth.

Both turret morphology and micromorphology were characterized over the three days of the rebuilding process via observations under a binocular and a petrographic microscope, respectively. For the latter, 30 µm thin sagittal sections of the turret walls were obtained, which included the wall surface facing the inner turret gallery. The morphology of the inner turret wall and its lining fabrics were described based on 12 original turrets and nine rebuilt turrets that were longitudinally sectioned to examine their inner surface. They were first slightly sprayed with hairspray prior to removal, and then fixed in the laboratory by carefully pouring an alcoholic solution of Butvar^®^ on them. For the micromorphological descriptions, 10 original turrets and 10 turrets rebuilt after either one, two, or three consecutive days, as well as samples of the surrounding soil, were impregnated with polyester resin to harden their structure without structural disturbances, and thin sections were obtained. All turrets were longitudinally sectioned except one original turret, 4.5 mm high, which was transversally sectioned to obtain a series of 14 thin sections taken every 3 mm from the base to the top, for additional comparisons.

Thin sections, 30 µm thick, were obtained following a methodology commonly used for micromorphological analysis of soils [38]. The petrographic microscope allowed those micromorphological features that could give information about the building techniques employed by ants to be characterized: for instance, the wall components and their distribution and assemblage to form different types of microstructures, the presence of wall linings, and structural beams. The sections were observed under crossed polarizers at 90°, which highlight and improve the observation of porous and poorly aggregated structures. The terminology employed to describe the microstructural types and micromorphological features was based on previous studies on soils, termites, and ants [5,14,20,39], but some new names are proposed herein to describe two microstructural types that have not been described yet (single and cemented fiber structures, see Section 3.1). The wall porosity was visually estimated as a proportion of void space in the thin section, and compared with graphics of abundance of black objects as a percentage of visual fields [40,41].

## 3. Results

### 3.1. Original Nest Turrets: External Morphology, Inner Walls, and Microstructures

Nests of the grass-cutting ant *A. fracticornis* are abundant and often located in open grasslands. Workers build a conspicuous cylindrical turret on the single nest entrance, often slightly bent laterally, typically made of grass fragments, which is not closed on a daily or seasonal basis. Figure 1 presents examples of turrets of unknown age from field nests, to show the diversity of their morphology.

Externally, some turrets appear to be made exclusively out of grass fragments, twigs, and leaf veins, while others include pellets of excavated earth materials. Due to the usually concentric deposition of the excavated materials around the single nest entrance, turrets are located at the top of the resulting elevation. Some turrets may have more than a single opening, up to five or six, which nevertheless converge at the soil surface in a single nest entrance. The colony’s waste dump is located beside the entrance and usually at the base of the earthen mound or a few cm away. Observations over the seasons indicated that nest turrets, if eroded or damaged, are more likely to be constructed or reinforced in the rainy season, as previously described for *A. landolti* [35], and no apparent turret-building activity occurs in the dry, winter months.

Turrets with a single opening may reach up to 7 cm without considering the earthen mound, but in average they were 2.84 cm high (±0.07 cm; *n* = 40), with the two largest perpendicular diameters at the base being 3.25 and 3.12 cm (±0.11 and ±0.14 cm, respectively), a diameter at the top of 1.97 cm (±0.06 cm), and an opening diameter of 0.94 cm (±0.04 cm). Regarding their material composition, which may depend on the season and availability of materials, turrets were composed of plant materials (35.6% ± 5.0% of total mass; mean ±SE; *n* = 5) and soil particles (64.4% ± 5.0%). Due to its lower specific mass, plant material represents the majority of the turret volume (an example is presented in the Appendix A).

Observations of original turrets of unknown age under the binocular microscope indicated that the turret walls were composed of intermeshed plant material, such as dry grass fragments, plant fibers, twigs and dry leaf fragments, cemented by soil aggregates (Figure 2). The proportion of vegetal fibers and soil material, as indicated above, varied from very solid turrets, mostly composed of soil, to mesh-like turrets mostly made out of plant material, and also varied from the turret base to the top.

Typically, plant fragments were tangentially distributed around the turret gallery, displaying a horizontal or angled orientation, rarely vertical. The soil aggregates, located between the plant fragments, were rounded pellets (0.7–1 mm in diameter), irregular shaped granules or layers, grey or reddish colored. They were randomly distributed, and their colors corresponded to those of the organic A and clayish AC horizons, respectively, indicating that ants removed soil from different depths at the time when they were lining the gallery. The gallery walls were generally plastered with soil linings composed of welded soil pellets. The linings looked like grey and pink “patchworks” of pellets randomly distributed. The lining fabrics showed three distinct patterns: open, intermediate, and close fabrics (Figure 2). In open and intermediate fabrics, soil pellets were located between the intermeshed plant materials and filled the spaces between them, being more abundant in the latter. In the close fabric, the plant fragments were completely hidden by the covering soil layer, showing a mamillated or a smooth texture. Close fabrics dominated at the base of the turret. At the middle region, all three fabrics were observed, whereas at the top, most turrets were not lined at all, and a few turrets showed variable lining fabrics (Figure 2a–c).

Regarding their micromorphological features of the turret wall, original turrets of unknown age were composed of soil material, very fine sands of quartz (50–100 µm of grain diameter), silt, and clay, and plant fibers, thin twigs, and leaves, mostly grasses. A precise mineralogical description of the incorporated soil materials was beyond the scope of the study and therefore not intended. Soil materials and plant fragments were variably distributed forming four different microstructural types (Figure 2), as follows. (1) The single fiber structure, a new microstructural type revealed here, is a net-like structure only composed of intermeshed vegetal fibers. Frequently, their surface shows scattered soil granules adhered. (2) In the cemented fiber structure, also a novel microstructural type shown here, small soil aggregates (granules or thin layers) connect the vegetal fibers forming a very porous structure (wall porosity ≥ 50%). (3) The pelletal structure displays a similar porosity (≥50%), but it is composed of welded rounded soil aggregates (pellets) forming mamillated masses frequently crossed by vegetal fibers. (4) The spongy structure combines both mineral and organic soil materials forming a less porous structure (wall porosity 20–50%), showing many interconnected and irregular-shaped pores. In the original turrets, a spongy structure was dominant at their base, whereas pelletal and cemented fiber structures were commonly observed in middle and top regions (Figure 2d–f), with single fiber and spongy structures being less common.

Gallery linings were recognized as soil coatings 200–500 µm thick, composed of very fine sands, silt, and clay. Their inner surface could be smooth or slightly undulated, but most linings showed a mamillated surface or were moniliform coatings composed of welded soil pellets. Open fabrics with the occurrence of only few pellets were often observed in the uppermost internal wall of the turret (Figure 3a), while close fabrics with a continuous pelletal lining mostly occurred at the turret base. (Figure 3b). Longitudinal thin sections evinced the occurrence of both undulated (Figure 3c) and wider, compact linings (Figure 3d).

### 3.2. Rebuilt Turrets: Dynamics of Inner Wall Lining Fabrics

The dynamics of the most common inner wall lining fabrics at different regions, for 1-, 2-, and 3-day-old turrets is presented in Figure 4. In 1-day-old turrets, the gallery was often not plastered (open fabric), so the initial mesh-like wall structure could still be observed. 

Open fabrics dominated the top region of 1-day-old turrets (Figure 4a). In fact, one third of all investigated turrets had galleries with open fabrics and no lining at the top, irrespective of their age. In the middle region of 1-day-old-turrets, open fabrics with scattered soil pellets between the mesh of plant fragments were the most common (Figure 4b), followed by intermediate fabrics. Intermediate fabrics displaying a continuous pelletal lining were observed at the base of most 1-day-old turrets (Figure 4c). A similar transition between the fabrics from the top to the base was also observed in 2-day-old turrets (Figure 4d–f). In 3-day-old turrets, galleries showed open fabrics at the top region (Figure 4g), as in younger turrets, and also close fabrics, which were still dominant in the middle region (Figure 4h). At the base, close fabrics forming a continuous pelletal lining that masked the whole mesh of plant fragments were dominant (Figure 4i). Even though no systematic quantification of turret heights and widths over time was made, it appeared that most turrets reached their final external size after one day, with only a few slightly growing during the second day. Taken together, the general trend was a change in the fabrics from open to close downwards along the turret structure, and over time. The above descriptions refer to the most common lining fabrics. Detailed information about the frequency of the less common lining fabrics is presented in the Appendix A.

### 3.3. Rebuilt Turrets: Dynamics of Inner Wall Micromorphology

The most common microstructural types observed at different regions of 1-, 2-, and 3-day-old turrets are comparatively presented in Figure 5.

At the top, 1-day-old turrets showed pelletal or single fiber structures (Figure 5a), which also occurred at the middle region, followed in frequency of occurrence by cemented fiber structures (Figure 5b). At the base, masses of welded soil pellets cementing the plant fragments formed a pelletal structure that prevailed in 1-day-old turrets (Figure 5c). Two-day-old turrets presented a similar pattern as 1-day-old turrets at the top and middle regions (Figure 5d,e), yet at the base, most 2-day-old-turrets showed mineral and organic soil materials exhibiting a spongy structure, with abundant interconnected voids (Figure 5f). Most 3-day-old turrets exhibited a pelletal structure followed by single and cemented fiber structures at the top (Figure 5g), a pelletal structure at the middle region (Figure 5h), and a spongy structure at the base (Figure 5i), as observed in 2-day-old turrets. Taken together, the general trend was a change in the inner wall micromorphology from single fiber structures, over cemented fiber and pelletal structures, to spongy structures downwards along the turret, and over time. The above descriptions refer to the most common wall micromorphological features observed in the thin sections. Detailed information about the frequency of the less common micromorphological features is presented in the Appendix A.

In addition to the analysis of the longitudinal thin sections of the turrets, as described above (Figure 5), we obtained a series of 14 transversal sections from a single turret of 4.5 cm in height, made every 3 mm from the base to the top. The observations confirmed the pattern described above: the occurrence of distinct micromorphological features at the base (0–9 mm in height), middle (>9–36 mm in height), and top (>36–45 mm in height) of the turret. Basal sections, taken from the ground level up to 9 mm, showed walls with a cemented fiber, pelletal, or spongy structure, crossed by particularly large fragments named “beams” (see below). The gallery had a smooth, undulated surface when the basal wall showed a spongy structure, or a distinct lining when the base showed a cemented fiber or pelletal structure. Middle sections also showed cemented fiber and pelletal structures and an undulated gallery lining. Sections from the upper part of the turret showed walls with a pelletal fiber structure and a discontinuous gallery lining. No beams were observed at the middle and top sections.

Samples of undisturbed soil collected below turrets located on grass-covered soils displayed a spongy structure with abundant vegetal material, whereas a compact microporous structure (porosity ≤ 20%) was observed in soil samples taken below turrets located on exposed, partially eroded soils. In the latter case, the soil was only composed of mineral grains densely cemented by fine material (clay and fine organic matter), showing numerous but very small pores (≤100 μm). In both soil samples, mineral grains were very fine with fine quartz sands (50–100 µm in diameter).

### 3.4. Onset of Turret Building: Use of Beams

Observations on the building behavior of ants after removal of the original turret suggested that workers initially pile up material at the exposed nest entrance, and often place large, more stable plant fragments partially covering or crossing the exposed nest entrance (Figure 6a,b), which may reinforce the base of the emerging structure.

We coined “beams”, as in human buildings, those large and bulky plant fragments, equal or wider than 1 mm, included in the turrets’ walls. In the thin sections, ovoid, empty voids of similar size were considered traces of beams. Most beams were tangentially located to the gallery, and a few showed a parallel orientation (Figure 6c,d). Beams often occurred at the base of the turret, and were also added over time at the middle and less at the top of the structure. After three days, the distribution of beams in the reconstructed turret was similar to that of the original turrets (Figure 6e).

## 4. Discussion

Workers of the grass-cutting ant *A. fracticornis* construct a turret with plant fragments and soil on top of their nest entrance. Our field experiments allowed a precise characterization of the most and less common morphological and micromorphological changes in the turret structure during its construction. Both revealed the involved techniques, showing that turrets do not result from solely piling up neighboring materials by workers, but represent the outcome of dynamic building behaviors. During the process of turret construction, workers display three concurrent building behaviors, namely: (1) intermeshing grass fragments and twigs around the nest opening, (2) cementing the resulting mesh with soil pellets, and (3) lining the inner gallery wall, as summarized in the following paragraphs.

### 4.1. Intermeshing Grass Fragments and Twigs around the Nest Opening

Grass fragments, mostly dry, and thin twigs are collected and carried by workers from the surroundings of the nest opening. At the beginning of the turret construction, they are horizontally placed across and surrounding the nest opening, and then piled and intermeshed, giving rise to a net-like, “basketry”, single fiber structure. The nest entrance is likely kept free over the building process because of the subsequent displacement and removal of those fragments that obstruct the ant traffic. Thicker sticks are generally located at the base and middle parts of the emerging turret and function as beams supporting the upper wall structure, i.e., they are incorporated into the turret structure at the very early stages of the construction.

### 4.2. Cementing the Vegetal Mesh with Soil Pellets

Concomitantly with the accumulation of plant fragments around the entrance, soil pellets are placed in between the intermeshed plant fragments, starting from the base of the turret towards the top, thus reinforcing the structure. Pellets arise from the underground excavation of the nest, because pellet-loaded workers were observed to come from the nest during turret construction, and pellets showed the same mineral composition as observed in the soil horizon where ants excavated their underground tunnels and chambers. Their rounded shape is conserved and can be recognized in those turret walls that show a pelletal structure. Generally, pellets are welded, so that their original rounded shape may change during cementation because of both the plasticity of the moist soil and the manipulation/pressing of workers during their incorporation into the structure, thus forming irregular or layered aggregates between the plant fibers.

The construction of a complete turret lasted about three days. Turrets developed rapidly during the first day, when the initial mesh of plant fragments was gradually cemented with soil pellets starting from the base upwards. During the second day, the turret cementation continued and around the third day, turrets displayed the most structurally stable condition: most turrets showed a spongy structure at the base, a pelletal structure in the middle region, and cemented fiber or single fiber structures at the top.

### 4.3. Lining the Inner Gallery Wall with Soil Pellets

Alongside with the cementation of the wall, the inner gallery was gradually lined with soil pellets to turn its initial irregular wall to a smooth surface that allows undisturbed ant traffic. Soil pellets were firstly placed between the plant fragments forming an open fabric; the pores were then gradually filled with soil pellets forming an intermediate fabric, resulting in a final close fabric where the whole surface was coated by a continuous pelletal lining. Close linings generally showed an undulated texture because of their pelletal origin, but in older turrets, the texture was smooth, probably because of traffic-dependent erosion or additional reworking of the surface by ants. Three-day-old turrets showed similar wall microstructures and lining fabrics to most original turrets, thus indicating that turret development indeed finished after about three days. Even though original turrets were randomly selected in field nests, and their age was unknown, most of them showed solid walls and looked like completed constructions.

### 4.4. Comparative Aspects of Turret Building

Turret construction occurs in rainy periods, when the soil has enough moisture and plasticity to be excavated and molded into pellets, and then employed to cement the initial mesh-like structure. In the rainy season, workers of *A. landolti* construct turrets with plant fragments densely intermeshed [30]. Navarro and Jaffé [35] tested the water permeability of *A. landolti* nest turrets and observed that their walls resisted infiltration for 15–90 min, and that their structure did not collapse even if some water infiltration occurred afterwards. Accordingly, turrets likely prevented water infiltration into the nest, since rain showers in that area very rarely exceed 60 consecutive minutes.

*A. landolti*, *A. balzani*, and *A. fracticornis*, the species of the present study, are phylogenetically closely related grass-cutting ant species [42,43] and they probably share very similar behaviors for the building of nest turrets with intermeshed grass fragments (although the use of soil as building material was not reported for *A. landolti* [35]). The construction of a single solid turret around the nest entrance is also known for the Ponerine ant *Ectatomma opaciventre*, and is composed of soil material cementing an internal structure of intermeshed dried plant fragments. Saliva proteins occur in the gallery lining [36], suggesting that their presence could harden and make the walls more resistant to water infiltration [37].

At the same field site where we performed the experiments on *A. fracticornis*, neighboring colonies of the leaf-cutting ant *Atta vollenweideri* construct conspicuous, large ventilation turrets with excavated soil pellets and plant fragments imported from the environment on the topmost openings of their huge mounds. Their walls show a very porous microstructure composed of welded soil pellets and abundant plant fragments [20]. The wall exhibits a similar pelletal microstructure to that observed in turrets of *A. fracticornis*, and the mineral composition also corresponds to the excavated soil horizon. Interestingly, nest turrets of *A. vollenweideri* show two micromorphological differences as compared to the turrets of *A. fracticornis*: firstly, no linings covered the inner gallery of *Atta* turrets, and secondly, the plant fragments included in the pelletal wall did not show any specific spatial orientation in relation to the opening [20]. Thus, we conclude that workers of *A. vollenweideri* construct the nest turrets by simply piling and welding excavated soil pellets, and by intermeshing plant fragments without any orientation to reinforce the wall structure, whereas in *A. fracticornis*, workers use soil pellets to line and cement grass fragments and twigs previously intermeshed around the nest entrance, forming a basketry structure. Interestingly, *Atta* workers are also able to line and reinforce unstable turret walls when needed: they were observed to line the gallery surface of mechanically unstable turrets constructed in the laboratory with coarse sands, thus reinforcing the fragile and breakable sandy structure with compact clay coatings [14].

The turrets of *A. vollenweideri* nests facilitate air exchanges between the nest and the environment via wind-induced, passive flows [19,44]. In the laboratory, it was demonstrated that CO_2_ levels in the outflowing nest air influenced turret construction: elevated CO_2_ levels led to the construction of a turret with a larger total aperture and several minor openings, which might facilitate the removal of CO_2_ from the underground nest structure [21]. The function of the *A. fracticornis* nest turrets is unknown. It is tempting to speculate that nest turrets of *A. fracticornis*, particularly those with more than one opening, may facilitate the renewal of nest air. However, it appears unlikely that the fungus gardens are exposed to detrimental CO_2_ levels because of their relatively shallow location in the soil as compared to the much deeper *Atta* nests. Nest turrets of the related grass-cutting ant species *Acromyrmex balzani* have been recently suggested to provide visual cues for returning workers [31]. Whether the *A. fracticornis* nest turrets facilitate air exchanges, prevent humidity losses, block water infiltration during floods, preclude the plugging of the nest entrance by soil particles blown by the wind, or help homing ants to visually locate the nest, remain to be explored in future studies.

Interestingly, turret building behavior has also been reported for the lower attine, fungus-growing ant *Cyphomyrmex longiscapus*, yet the turrets display a very different morphology and their function is unknown [45]. It is a forest-dwelling species and nests occur on steep embankments along streams in wet forests, being protected from heavy rains underneath overhangs or in crevices. Nests are composed of a shallow single cavity containing the fungus, with a characteristic entrance surrounded by a funnel-shaped turret. The turret and parts of the cavity walls, especially those supporting the entrance auricle, are constructed by layering clay crumbs. Even though the broad entrance is partly closed by a clayish “trabeculae”, the fungus garden, situated immediately behind the trabeculae, is partly exposed to the environment. This exposure of the garden is atypical for attine ants and may explain why *C. longiscapus* occurs in wet habitats where the fungus is not prone to desiccation [45]. The entrance turret may offer some mechanical protection to the partly exposed fungus garden, yet its precise function remains unexplored. 

Construction of nest turrets is not only limited to societies of fungus-growing ants. Solitary wasps of the species *Odynerus dilectus* and *Paralastor* sp. (Eumeninae, Vespidae) also construct turrets with moist soil pellets above their subterranean cells that look like muddy turrets of *A. fracticornis* with pelletal structure. Though their function was not demonstrated, it was proposed that they prevent the plugging of the nest opening by wind-blown soil or prevent humidity losses from the nest [46,47]. The “turret spider” *Tarentula arenicola*, as indicated by its name, surmounts its underground nest with a turret composed of soil material and intermeshed twigs, with its upper part lined with silk. In sandy areas, the complete turret structure is held by threads of silk, indicating that the spider responds to the mechanical instability of the sandy structure and reinforces the emerging turret accordingly [48], as mentioned above for *Atta vollenweideri* leaf-cutting ants [14].

*Tarentula* spiders, as well as *Odynerus* and *Paralastor* wasps, are solitary builders that display, with flexibility, all necessary building behaviors to construct a nest turret. In *Acromyrmex* grass-cutting ants, the three main building behaviors displayed during turret construction, namely, intermeshing plant fragments around the entrance, cementing them with soil pellets, and lining the inner gallery wall, occur concurrently and need to be coordinated in time and space over the building process. Ants collectively construct their nest turret following engineering principles that, as an analogy, human builders adhere to in much the same way when building a wooden frame house, as follows: first the stabilizing base, then the wooden frames, then the sheet rock, and finally the plaster. It remains a challenge to understand how such a sophisticated coordination during turret building in grass-cutting ants is achieved. For instance, whether communication signals are involved (as in digging behavior by *Atta* workers; [16]), and whether there are building specialists among workers that only perform particular tasks, or individual ants that change their building tasks as a response to specific, new stimuli provided by the emerging turret structure over time.

## 5. Conclusions

Grass-cutting ants, *Acromyrmex fracticornis*, construct conspicuous nest turrets made of interlaced plant fragments. We asked whether turrets are built by merely piling up nearby materials around the nest entrance, or whether ants incorporate different materials as the turret develops. Our experiments revealed that consolidation of the turret occurred both over time and from its base upwards. Different building behaviors are involved, namely (1) intermeshing grass fragments and twigs around the nest opening, (2) cementing the resulting mesh with soil pellets, and (3) lining the inner gallery wall. Nest turrets do not simply arise by the arbitrary deposition of nearby materials, since workers both selectively incorporate large materials at the beginning, and respond to the developing structure by reinforcing the intermeshed plant fragments over time.

## Figures and Tables

**Figure 1 insects-11-00140-f001:**
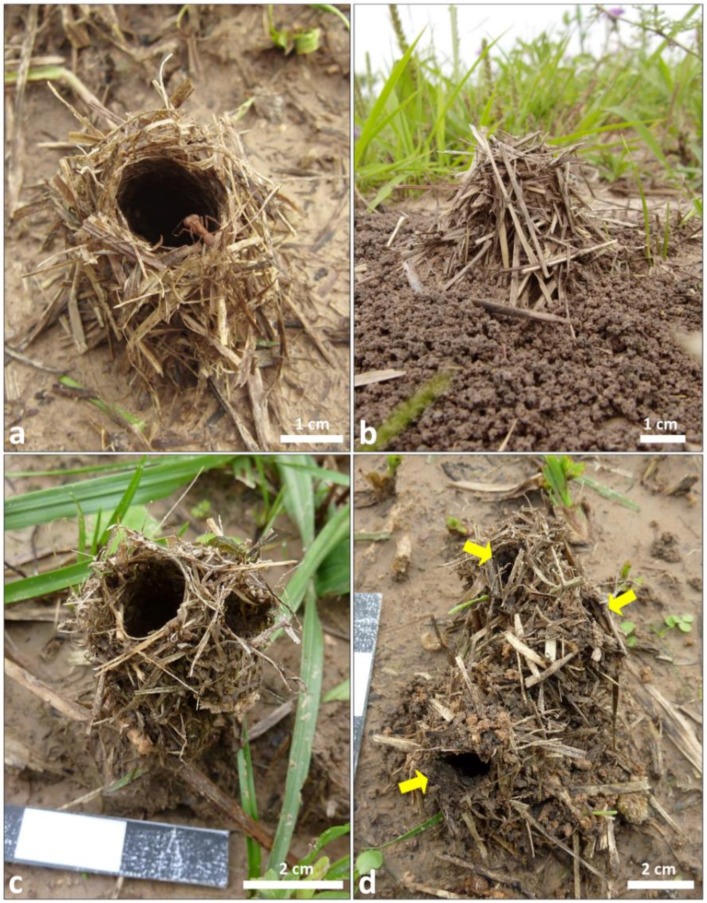
Examples of turrets of unknown age from field nests of *Acromyrmex fracticornis* located in Formosa, Argentina. (**a**) Turret built with grass fragments, internally lined with soil pellets (not visible). (**b**) Lateral view of a turret built with grass fragments located on top of a slightly elevated mound composed of excavated soil pellets. (**c**) Example of a turret with two adjacent openings that merged and led to a single nest entrance at the soil level (not visible). (**d**) Example of a turret with three openings (yellow arrows). Excavated soil pellets are located in between the grass fragments. Photo credits: (**a**,**b**): F. Roces; (**c**,**d**): D. Römer.

**Figure 2 insects-11-00140-f002:**
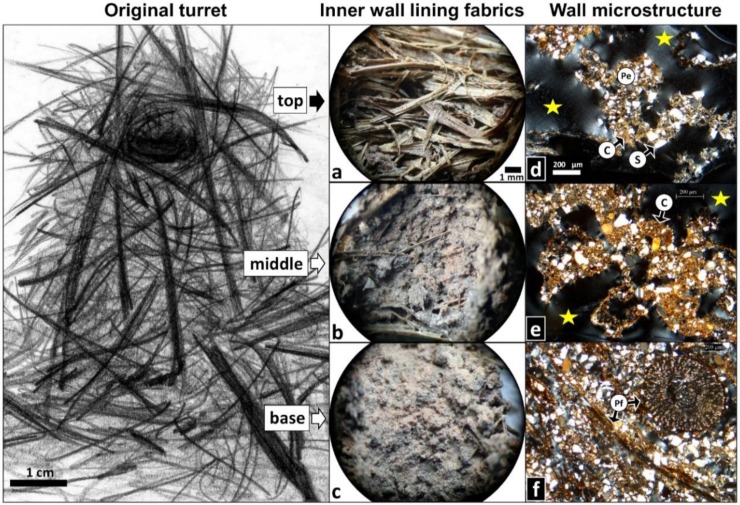
Morphology of original turrets of unknown age collected from field nests (schematic drawing of an entire turret by M. I. Cosarinsky). Shown are the most common lining fabrics found on the inner wall at the turret top, middle, and base, as well as the most common wall microstructures identified on thin sections. Inner wall lining fabrics: (**a**) not lined (*n* = 5/9, i.e., found in five out of nine turrets). (**b**) Open and intermediate lining fabrics (*n* = 7/9). (**c**) Close fabric (*n* = 7/9). Micromorphology of the turret wall: (**d**) pelletal fiber structure (*n* = 7/10). Highlighted is a pellet (Pe) composed of sand grains (S) of variable size cemented with bright clay (C), i.e., birefringent under polarized light, which may indicate that the clay aggregate contained no or only very few organic materials. Voids are indicated with yellow stars. (**e**) Cemented fiber structure (*n* = 8/10). The highlighted, darker clay is likely mixed with organic material. (**f**) Spongy structure (*n* = 6/10). Highlighted are plant fragments (Pf) embedded in the structure, one of them transversally sectioned. For detailed descriptions of linings and microstructures see text. Scale bar in (**a**) is also valid for (**b**) and (**c**); scale bar in (**d**) is also valid for (**e**) and (**f**). Photo credits: M. I. Cosarinsky.

**Figure 3 insects-11-00140-f003:**
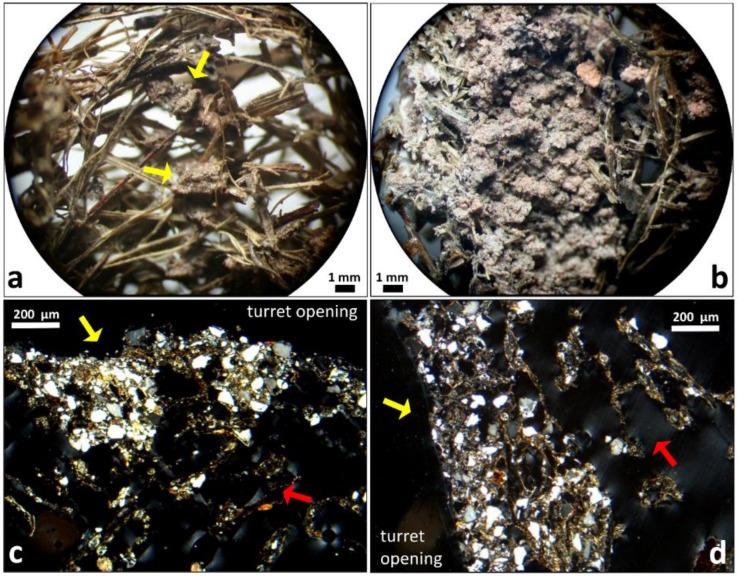
Examples of lining fabrics of the internal turret wall. (**a**) Open fabric as observed in the uppermost internal part of the turret, with the occurrence of a few soil pellets (yellow arrows) lined between vegetal fibers. (**b**) Close fabric with a continuous pelletal lining covering all fibers of the internal wall, as observed at the turret base. (**c**) Thin section showing an example of an undulated lining (yellow arrow) on the inside wall of the turret and the location of the turret opening. Two aggregates of sand grains cemented with clay are visible. The red arrow points at the porous turret wall. (**d**) Thin section showing a wider, compact, and continuous lining (yellow arrow) on the inside wall of the turret opening, and the adjacent turret wall (red arrow). Photo credits: M. I. Cosarinsky.

**Figure 4 insects-11-00140-f004:**
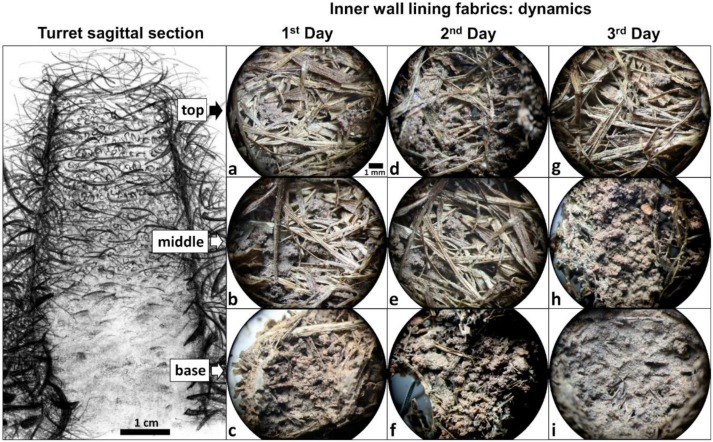
Building dynamics of a new turret after complete removal of the original one, over three days. Shown are the most common lining fabrics found on the inner wall at the turret top, middle, and base (schematic drawing of a sagittal section of an entire turret by M. I. Cosarinsky). (**a**,**b**) Scattered soil pellets between intermeshed plant fragments displaying an open fabric (*n* = 5/9 and *n* = 6/9, respectively). (**c**) Discontinuous pelletal lining composed of soil pellets welding plant fragments, displaying an intermediate fabric (*n* = 6/9). (**d**,**e**) Open fabrics (*n* = 5/9 each). (**f**) Intermediate fabrics (*n* = 6/9). (**g**) Open fabric (*n* = 3/9). (**h**,**i**) Continuous pelletal lining masking the plant fragments, displaying a close fabric (*n* = 6/9 and *n* = 7/9, respectively). Scale bar in (**a**) is valid for all photos. The frequency of occurrence of the less common lining fabrics is presented in the Appendix A. Photo credits: M. I. Cosarinsky.

**Figure 5 insects-11-00140-f005:**
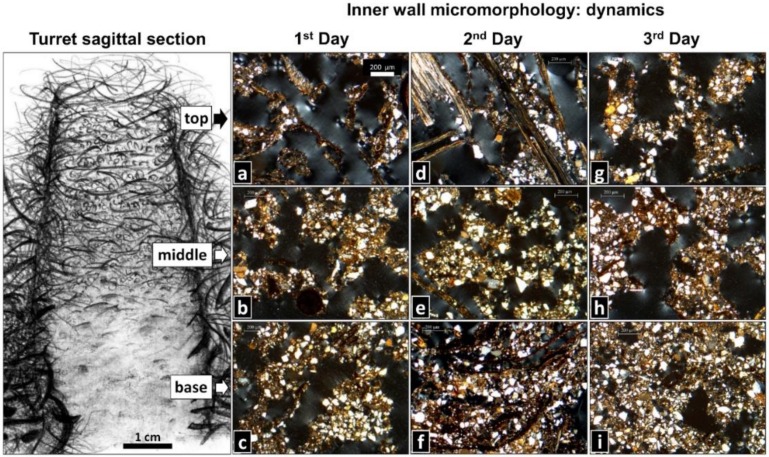
Building dynamics of a new turret after complete removal of the original one, over three days. Shown are the most common wall microstructures identified on thin sections (schematic drawing of a sagittal section of an entire turret by M. I. Cosarinsky). (**a**) Intermeshed plant fragments displaying a single fiber structure (*n* = 5/10). (**b**,**c**) Masses of welded soil pellets cementing plant fragments, displaying a pelletal structure (*n* = 6/10 and *n* = 8/10, respectively). (**d**) Single fiber structure (*n* = 5/10). (**e**) Pelletal structure (*n* = 5/10). (**f**) Mineral and organic soil materials with abundant interconnected voids, displaying a spongy structure (*n* = 5/10). (**g**,**h**) Pelletal structure (*n* = 5/10 and *n* = 7/10, respectively). (**i**) Spongy structure (*n* = 7/10). Scale bar in (**a**) is valid for all thin sections. The frequency of occurrence of the less common inner wall micromorphological features is presented in the Appendix A. Photo credits: M. I. Cosarinsky.

**Figure 6 insects-11-00140-f006:**
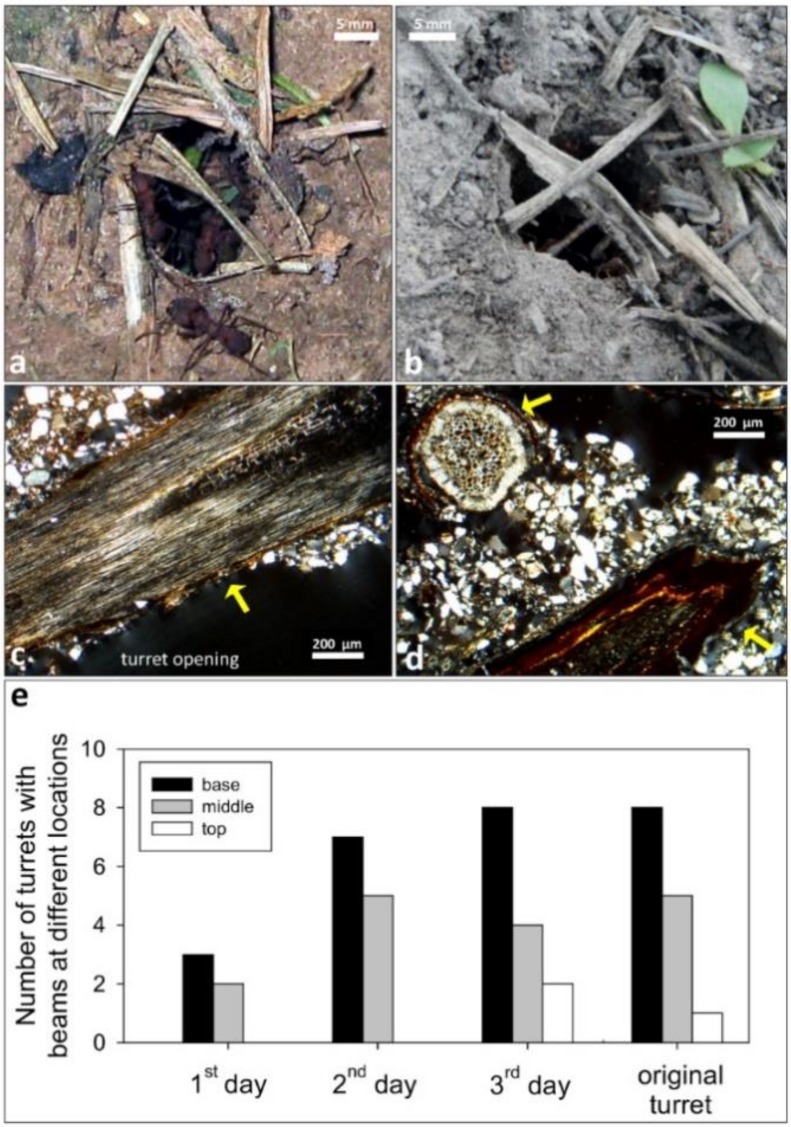
Examples of the occurrence of beams in the turret structure. (**a**) Photo of a nest entrance after removal of the original turret, to show the placement of a beam across the opening. (**b**) An additional example showing the position of the two first beams carried and placed by ants on the free nest entrance. (**c**,**d**) Two thin sections of different turret walls to illustrate the occurrence of beams—yellow arrows—embedded in the turret wall. In (**c**), the large beam was facing the turret opening and partially coated with sand grains and clay aggregates. (**e**) Occurrence of beams at different locations in the structure of the turrets rebuilt over three days, as compared to the original turrets. No attempt was made to perform statistical comparisons of the counts, because of the low sample size (*n* = 10 turrets analyzed in each group). Photo credits: (**a**): D. Römer; (**b**): F. Roces; (**c**,**d**): M. I. Cosarinsky.

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
