# Peer review of "Nest Turrets of Acromyrmex Grass-Cutting Ants: Micromorphology Reveals Building Techniques and Construction Dynamics"

_insects, 2020, doi:10.3390/insects11020140_

Round 1

Reviewer 1 Report

Thank you for preparing such an excellent manuscript, it was a joy to read and a brilliant contribution to the field. 

Having read it thoroughly, I can suggest no major or minor corrections. My only possible thought was to suggest that one additional possible function of the turret could be as a form of mimicry, to resemble the turrets of other more menacing critters. 

Your work would find a receptive audience in high profile journals such as  J. Roy. Soc. Interface, where a mix of biologists and engineers would enjoy speculating about the biomechanical nature of these turrets and their self-organized construction.

Author Response

Please see the attached word file containing all reviewers' questions and our responses.

Reviewer 2 Report

The manuscript is well written and presents result of Acromyrmex fracticornis grass-cutting ants. This research has a hight originality anf qualification to be published in INSECTS

Author Response

Please see the attached word file containing all reviewers' questions and our replies.

Reviewer 3 Report

This manuscript explores the construction methods for nest turrets in the ant species Acromyrmex fracticornis. The authors removed existing turrets in the field to take measurements and to analyse the structure. They also followed the reconstruction of the turrets for 3 consecutive days, collecting part of them on each day. They concluded that the turrets are built up successional, first with plant material, then clay pellets are added and finally the inner wall, especially in the bottom is lined.

This is an interesting and well written study and I only have few comments that could improve the manuscript which I will line out below. I recommend the editor to accept this manuscript after minor revision.

Comments 

Results

I think the results section would benefit from a division with section headers. I find it at times difficult to understand what part of the experiment is talked about. There are a few sentences that start with e.g. “Figure 4 compares…” which in my opinion isn’t very elegant. When section headers are added this type of sentences could be avoided.

I was wondering if the authors could give a bit more explanation regarding the figures. For someone like me with not much experiences seeing this kind of photographs, i.e. the micromorphology photographs, it is hard to interpret what is to see. E.g., what are the white grains that can be seen? What are the dark brown parts and what the light brown parts? I suggest the authors explain this upon first occurrence of the photographs.

L157-158 “The colony dump … a few cm away.”

It is unclear to me what the authors mean by the colony dump in this sentence. Do they mean the waste/debris dump of coming from the fungus gardens? Please, clarify this.

L250-252 “Taken together … common lining fabrics.”

I was wondering if the authors also looked at the size changes over time. Are the turret heights and widths different between days 1, 2 and 3? This information could give extra information on turret information. If the authors have these data available, or are able to obtain this, I think it would add value to the manuscript.

L310-311 “No attempt was … low sample size”

I appreciate the authors left out the statistical analyses for this, but I think it would be better if at least the error bars were shown for the bargraph. This would show the readers the variation of the data.

Discussion

L328 “but represent the final outcome”

I suggest the authors to remove the word ‘final’, because outcomes are final.

Author Response

(The authors gave the same response as above.)

Reviewer 4 Report

            This is the kind of thorough, detailed study I have come to expect from the Roces research group.  The study describes the structure of the turrets, including the less frequent structures, and follows the dynamics of how these turrets are built by the ants. The drawings by Cosarinsky are helpful in placing the more detailed images and descriptions into the whole-turret context.

            My sole suggestion for improvement is that the Discussion largely repeats the results, with a few added details and observations.  This is not necessary, as the Results section has already hammered all the points home several times.  I suggest removing these results from the Discussion and incorporating any new details in the Results.

            What I liked about this paper is that it adds another example of how sophisticated and competent ants are.  The authors don't even mention this more charming aspect of their discovery.  The ants adhere to engineering principles in much the same way that human builders do.  First the base plate, then the two-by-four frame, then the sheet rock, then the plaster and paint.

            The references are all numbered twice.

Author Response

(The authors gave the same response as above.)
